# Comparison of Temperature and Pain Changes between the Drip and Topical Methods of Administering the Transnasal Sphenopalatine Ganglion Block

**DOI:** 10.3390/jpm12050830

**Published:** 2022-05-19

**Authors:** Na Eun Kim, Ji Eun Kim, Sook Young Lee, Ho Young Gil, Sang Kee Min, Bumhee Park, Seung Il Kim, Ra Yoon Cho, Jae Chul Koh, Yi Hwa Choi, Jae Hyung Kim, Sang Jun Park, Jong Bum Choi

**Affiliations:** 1Department of Anesthesiology and Pain Medicine, Inha University School of Medicine, Incheon 22212, Korea; friskygirl@naver.com; 2Department of Anesthesiology and Pain Medicine, Ajou University School of Medicine, Suwon 16499, Korea; beye98@aumc.ac.kr (J.E.K.); anesylee@aumc.ac.kr (S.Y.L.); kilhoyoung@aumc.ac.kr (H.Y.G.); anesmin@aumc.ac.kr (S.K.M.); orange07@aumc.ac.kr (R.Y.C.); 3Department of Biomedical Informatics, Ajou University School of Medicine, Suwon 16499, Korea; bhpark@aumc.kr; 4Office of Biostatistics, Medical Research Collaborating Center, Ajou Research Institute for Innovative Medicine, Ajou University Medical Center, Suwon 16499, Korea; kimseungil@aumc.ac.kr; 5Department of Anesthesiology and Pain Medicine, Korea University School of Medicine, Anam Hospital, Seoul 02841, Korea; jaykoh@naver.com; 6Department of Anesthesiology and Pain Medicine, Hallym University School of Medicine, Hallym Sacred Heart Hospital, Anyang 14068, Korea; pcyhchoi@hallym.or.kr; 7Department of Anesthesiology and Pain Medicine, Hallym University College of Medicine, Hallym Dongtan Sacred Heart Hospital, Hwaseong 18450, Korea; jaehkim@hallym.or.kr; 8Department of Anesthesiology and Pain Medicine, Yonsei University School of Medicine, Severance Hospital, Seoul 03722, Korea; iotas@naver.com

**Keywords:** sphenopalatine ganglion, sphenopalatine ganglion block, headache, herpes zoster, migraine, visual analogue scale

## Abstract

The objective of this study was to compare facial temperatures and the visual analogue scale (VAS) between the drip method and the topical method of transnasal sphenopalatine ganglion block (SPGB). The transnasal SPGB is administered to patients with facial or head and neck pain. In the transnasal approach, the drip and topical methods are frequently used. We compared facial temperatures and VAS after transnasal SPGB. Medical records of 74 patients who visited the pain clinic and underwent transnasal SPGB were retrospectively reviewed. A total of 156 transnasal SPGB were performed. The patients were divided into the drip-method and topical-method groups. Facial temperatures were measured in six areas of the right and left forehead, maxilla, and mandible before and 30 min after completion of the transnasal SPGB. Temperatures were compared before and 30 min after SPGB in each group and between the two groups. VAS scores were compared at the same times of SPGB in each group and between the two groups. In the drip-method group, there were significant increases at four areas of the face in temperature changes at 30 min after SPGB. In the topical-method group, there was no significant difference in the temperature changes at 30 min after SPGB. There were statistically significant differences in the facial temperature changes between the two groups in the right forehead (*p* = 0.001), left forehead (*p* = 0.015), and right maxillary area (*p* = 0.046). In herpes zoster, there were statistically significant differences in the VAS scores between before and 30 min after SPGB in both groups (*p* < 0.001, *p* = 0.008) and between two groups (*p* < 0.001). In migraine, there were statistically significant differences in VAS scores between before and 30 min after SPGB in both groups (*p* < 0.001, *p* = 0.004) and between two groups (*p* = 0.014). Transnasal SPGB using two methods showed different temperature changes and VAS scores.

## 1. Introduction

The sphenopalatine ganglion is located within the pterygomaxillary fissure and is surrounded by the palatine bone [1]. The sphenopalatine ganglion is composed of sympathetic, parasympathetic, and sensory components. Parasympathetic components are known to be dominant [2]. A sphenopalatine ganglion block (SPGB) is an interventional procedure used to treat head and neck pain. There are transnasal, transoral, subzygomatic, and lateral infratemporal approaches. The ganglion can be identified by injecting a radiopaque contrast medium. Further treatments, such as SPGB with local anesthetics, radiofrequency ablation, or stereotactic radiosurgery, should be decided after the identification of the sphenopalatine ganglion [3,4,5,6]. In the transnasal approach, several authors have described the traditional technique using sterile 10 cm cotton-tipped applicators that are dipped in the chosen anesthetic and then moved forward along the superior border of the middle turbinate, until reaching the posterior wall of the nasopharynx [7,8,9]. This is a transnasal topical method in which a cotton tip applicator soaked in a local anesthetic is inserted through the nostril and mounted onto the nasal pharynx, allowing the drug to reach the sphenopalatine ganglion by diffusion. Other techniques have been described by dripping 1 or 2 mL of the anesthetic into the nostril [2]. This includes a transnasal drip method that delivers local anesthetics through the nostril while the patient is lying down. Therefore, local anesthetics are delivered to the nasal pharynx and diffuse into the sphenopalatine ganglion and sympathetic chain (Figure 1).

## 2. Materials and Methods

This was a retrospective, case-control study. Reviews of medical records were approved by the institutional review board of Ajou University Hospital of Korea (IRB No. AJIRB-MED-MDB-20-565) and registered at ClinicalTrials.gov (Identifier: NCT04479176, https://clinicaltrials.gov/ct2/show/NCT04479176, accessed on 18 July 2020). The requirement for informed consent was waived because of the retrospective, case-control nature of the study. This research was performed at the Ajou University School of Medicine in Suwon, Republic of Korea. Medical records were retrieved from January 2019 to December 2019. Recruitment and data collection were performed from July 2020 to August 2020.

### 2.1. Participants

Based on the medical records, we retrospectively enrolled 74 patients who underwent transnasal SPGB. A total of 156 transnasal SPGB procedures were performed in 74 patients. The drip method was performed 111 times in 38 patients, and the topical method was performed 45 times in 36 patients. The inclusion criteria were as follows: (1) age over 20 years; (2) head and neck painful or unpainful diseases like herpes zoster, migraine with or without aura, trigeminal neuralgia, atypical facial pain, anosmia, tinnitus, and hyperhidrosis; (3) pain score > 4 on the VAS in the head and neck; and (4) having undergone a transnasal SPGB and having had facial temperatures measured. The exclusion criteria were as follows: (1) patients with a history of head and neck surgery, (2) treatment with a vasodilator or vasoconstrictor, (3) contraindication to treatment using a transnasal approach, (4) clinically significant systemic disease or any reduced organ failure, and (5) missing data. The patients were divided into the drip-method and topical-method groups. In both groups, facial temperatures were measured using skin thermometers (GE Marquette DASH 3000; GE Medical Systems Information Technologies, Inc., Milwaukee, WI, USA) with the patient in a supine position before and 30 min after the completion of the transnasal SPGB.

### 2.2. Procedure

#### 2.2.1. Transnasal SPGB by the Drip Method

Transnasal SPGB was performed by a single pain clinician. After the patient was placed in a supine and neck-extended position, 2 mL of 2% mepivacaine was placed in a syringe connected to a 16-gauge Angiocath sheath (Becton Dickinson Medical, Tuas, Singapore) (Figure 2A,B). The sheath of the Angiocath was inserted through the nostril, and 2% mepivacaine was dripped into the nostrils with the patient in a supine position (Figure 1A). The mepivacaine drip on the nasal pharynx was maintained for 10 min. A drip of 2% mepivacaine was delivered to the nostril, where the pain was dominant. In cases of bilateral pain, a mepivacaine drip was administered to both nostrils.

#### 2.2.2. Transnasal SPGB by the Topical Method

Transnasal SPGB was performed by a single pain clinician. The posture was the same as that in the drip method. A cotton tip applicator (Figure 2C) soaked with 2% mepivacaine was inserted vertically into the nostril (Figure 1B). After the cotton tip applicator made contact with the posterior wall of the middle turbinate, the cotton tip applicator was fixed for 10 min. A cotton tip applicator was inserted into the nostril, where the pain was dominant. In cases of bilateral pain, two applicators were inserted into both nostrils.

### 2.3. Clinical Evaluations

#### 2.3.1. Temperature Measurements

In the drip-method group, patients were laid down for 30 min. After waiting for 30 min in room air, skin thermometer probes (GE Marquette DASH 3000; GE Medical Systems Information Technologies, Inc., Milwaukee, WI, USA) were attached to six regions on the left and right sides of the forehead, maxillary area, and mandibular area of the face with the patient in a supine position (Figure 3). Facial temperatures were measured before transnasal SPGB and 30 min after the completion of transnasal SPGB.

In the topical-method group, temperature measurements were performed using the same method as in the drip-method group.

#### 2.3.2. Visual Analogue Scale

The VAS score was evaluated at the same time as the temperature measurements in herpes zoster and migraine patients.

### 2.4. Statistical Analysis

The sample size was 102 from a pilot study estimated using the data of patients who were previously assessed, which achieved a power of 80.427% and a mean difference of -0.37, with a standard deviation of 0.67 for both groups and a significance level (alpha) of 0.050. All data are presented as the mean and standard deviation. The paired *t*-test was used to compare the differences in temperatures before and after transnasal SPGB within groups, and independent two-sample *t*-tests were used to compare the differences in temperatures and VAS scores between the two groups. Statistical Analysis Software v. 9.4 (SAS Institute Inc., Cary, NC, USA) was used for all analyses.

## 3. Results

The demographic data are shown in Table 1 and the diagnoses are shown in Table 2. A flowchart of the research is shown in Figure 4.

In the drip-method group, facial temperatures increased in all six areas. There were significant differences at four areas on both sides of the maxillary and mandibular areas in the temperature changes before and 30 min after the drip method of transnasal SPGB (Table 3).

In the topical-method group, there was no significant difference in the temperature change before and 30 min after the topical method of transnasal SPGB (Table 3). There were statistically significant differences in the facial temperature changes between the two groups in the right forehead (*p* = 0.001), left forehead (*p* = 0.015), and right maxillary area (*p* = 0.046) (Table 4). In patients with herpes zoster, there were statistically significant differences in VAS scores between before and 30 min after transnasal SPGB with both drip and topical methods (*p* < 0.001, *p* = 0.008) and between two groups (*p* < 0.001) (Table 5). In patients with migraine, there were statistically significant differences in VAS scores between before and 30 min after transnasal SPGB with both drip and topical methods (*p* < 0.001, *p* = 0.004) and between two groups (*p* = 0.014) (Table 5).

## 4. Discussion

In this study, the drip-method group showed that the facial temperatures increased after transnasal SPGB, but the topical-method group showed no significant changes in facial temperatures after transnasal SPGB (Table 3). The changes in facial temperatures after transnasal SPGB were statistically different between the drip and topical-method groups (Table 4).

In a previous study, the drip method showed a temperature increase after a transnasal SPGB [10], while the topical method showed that the temperature decreased or remained unchanged after transnasal SPGB [11]. However, no studies have compared the changes in temperature between the drip and topical methods.

Transnasal SPGB was first introduced in the 1990s and has been widely implemented as a relatively non-invasive head and neck treatment to replace other injection treatments [3]. Recently, transnasal SPGB tools, such as Tx360^®^ (Tian Medical, Grayslake, IL, USA) and Sphenocath^®^ (Dolor Technologies, Scottsdale, AZ, USA), have been developed and used [2]. However, most of these dedicated tools adopt the drip method, indicating that facial temperatures have increased. Based on these results, it is considered that the sympathetic block is dominant in the drip method. However, in the topical method, since the facial temperature decreases or remains unchanged, it seems that the parasympathetic block is dominant or both the parasympathetic and sympathetic blocks are balanced. This seems plausible, even when considering the anatomical structure (Figure 1). Because sympathetic nerve fibers originate from the cervical sympathetic chains, the drip method may cause local anesthetics to fall into the pharynx, blocking the cervical sympathetic chain. Parasympathetic nerves originate from the facial nerve; therefore, the topical method is expected to block SPG which is dominant in the parasympathetic component, positioned in the upper area rather than the cervical sympathetic chain (Figure 1).

These different dominant effects of transnasal SPGB are important for the treatment of some diseases. It is widely known that sympathetic blocks are effective in herpes zoster or hyperhidrosis of the head and neck, and vasoconstrictive effects of parasympathetic nerve systems are theoretically expected to be effective in migraine or post-dural puncture headache (PDPH) [12,13]. Thus, the drip method, with a sympathetic-dominant block, may be effective in herpes zoster and hyperhidrosis cases, rather than in migraine and PDPH cases. Meanwhile, the topical method, with a parasympathetic-dominant or autonomic balanced block, may be effective in migraine and PDPH cases rather than in herpes zoster and hyperhidrosis cases (Table 6). However, VAS scores in herpes zoster and migraine decreased in both the drip and topical-method groups after transnasal SPGB and there were significant differences in VAS score changes between the drip method and topical method (Table 5). Since the two methods do not completely distinguish the sympathetic nerve block and the parasympathetic and sensory nerve block, the effects of these methods may converge, appearing increasingly similar over time. However, further research is needed on this topic.

Our findings need to be interpreted in light of the limitations of this study: (1) This study is retrospective; (2) the sample size was small; (3) this study was not a randomized controlled study; (4) this study included a heterogeneous disease group; (5) this study did not conduct a follow-up over a long-term period.

## 5. Conclusions

Transnasal SPGB using two methods (drip and topical) showed different temperature changes and VAS scores. The drip method showed increases in facial temperatures after SPGB, and the topical method did not show temperature changes statistically. Two methods of SPGB are expected to treat different diseases due to different effectiveness.

## Figures and Tables

**Figure 1 jpm-12-00830-f001:**
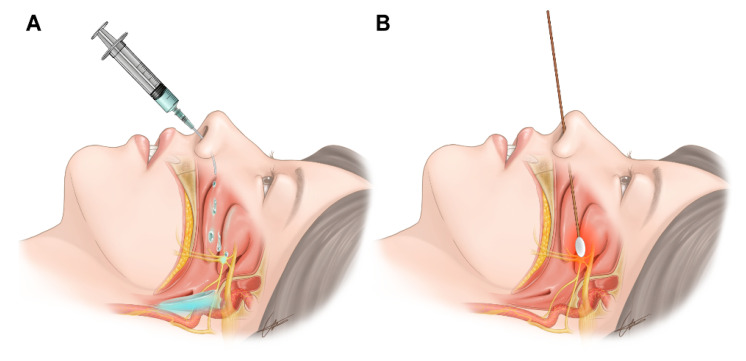
Transnasal sphenopalatine ganglion block by (**A**) the drip method and (**B**) topical method.

**Figure 2 jpm-12-00830-f002:**
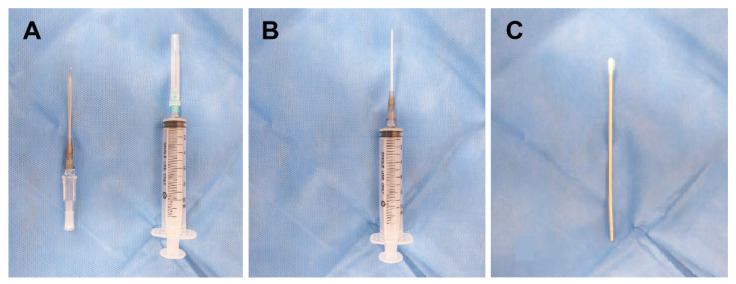
The tools for transnasal sphenopalatine ganglion block. (**A**) Angiocath, 10 mL syringe. (**B**) Syringe connected with Angiocath. (**C**) Cotton tip applicator.

**Figure 3 jpm-12-00830-f003:**
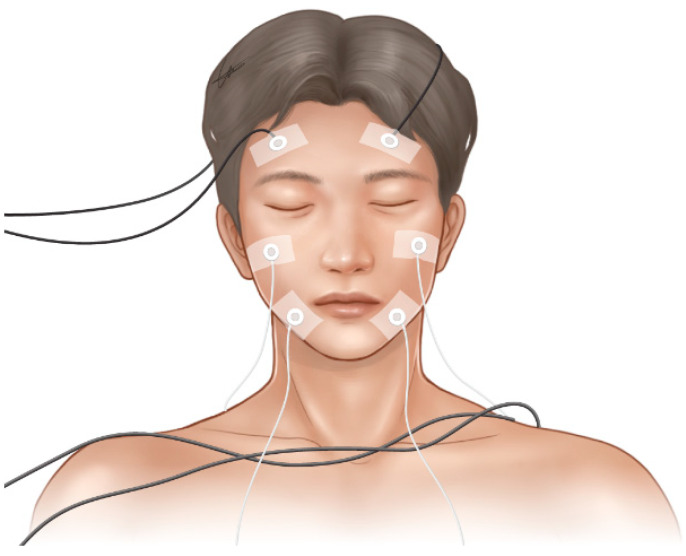
Measurement of facial temperature in six regions (right forehead area, left forehead area, right maxillary area, left maxillary area, right mandibular area, left mandibular area) by a skin thermometer.

**Figure 4 jpm-12-00830-f004:**
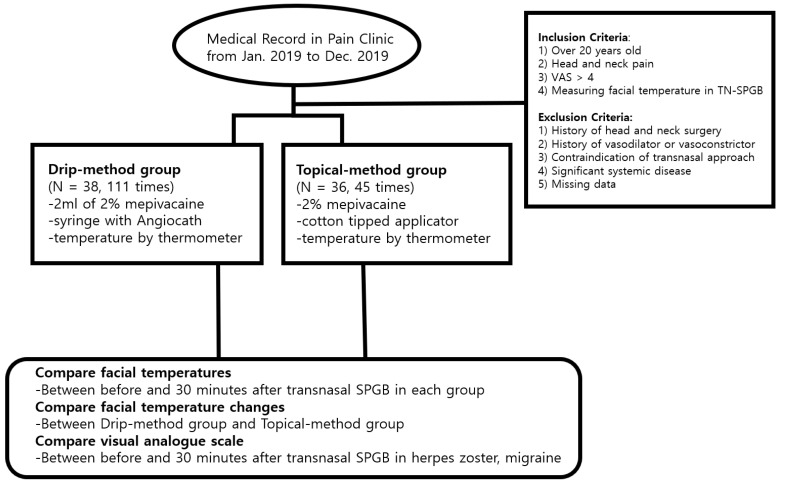
Flowchart of research. Visual analogue scale (VAS); sphenopalatine ganglion block (SPGB).

**Table 1 jpm-12-00830-t001:** Demographic data.

Parameters	Drip-Method Group (*n* = 38)	Topical-Method Group (*n* = 36)	*p*-Value
Age (years)	55.8 ± 13.2	58.6 ± 11.9	0.212
Sex (male/female)	15/23	17/19	0.557
Height (cm)	161.4 ± 7.2	162.0 ± 8.6	0.738
Weight (kg)	68.7 ± 10.6	67.2 ± 11.85	0.566

**Table 2 jpm-12-00830-t002:** Diagnoses.

Diagnosis	Drip-Method Group	Topical-Method Group	Total
Herpes zoster (*n*/times)	12/55	29/34	41/89
Migraine (*n*/times)	20/48	4/7	24/55
Hyperhidrosis (*n*/times)	1/1	-/-	1/1
Atypical facial pain (*n*/times)	4/5	1/1	5/6
Anosmia (*n*/times)	-/-	2/3	2/3
Tinnitus (*n*/times)	1/2	-/-	1/2
Total (*n*/times)	38/111	36/45	74/156

**Table 3 jpm-12-00830-t003:** Temperature measurements.

Methods	Regions	Temperatures (°C) (Mean ± SD)		*p*-Value
		Before Transnasal SPGB	30 min after Completion of the Transnasal SPGB	
Drip-method group	Right forehead area	33.49 ± 1.09	33.73 ± 1.00	0.095
	Left forehead area	33.31 ± 1.03	33.50 ± 1.06	0.174
	Right maxillary area	32.38 ± 1.76	32.98 ± 1.50	0.006 *
	Left maxillary area	32.28 ± 1.71	32.78 ± 1.53	0.025 *
	Right mandibular area	32.52 ± 1.52	32.95 ± 1.26	0.023 *
	Left mandibular area	32.49 ± 1.59	32.93 ± 1.36	0.025 *
	Total	32.75 ± 1.54	33.15 ± 1.34	<0.001 *
	Right forehead area	34.25 ± 0.94	34.08 ± 0.62	0.301
Topical-method group	Left forehead area	34.17 ± 1.00	34.02 ± 0.68	0.403
	Right maxillary area	33.84 ± 1.40	34.05 ± 1.01	0.415
	Left maxillary area	33.69 ± 1.40	34.01 ± 0.86	0.199
	Right mandibular area	34.09 ± 1.29	34.25 ± 0.77	0.476
	Left mandibular area	34.06 ± 1.24	34.22 ± 0.70	0.446
	Total	34.02 ± 1.23	34.10 ± 0.78	0.325

* *p*-values less than 0.05 indicate a statistical difference. SPGB, sphenopalatine ganglion block; SD, standard deviation.

**Table 4 jpm-12-00830-t004:** Comparison between drip and topical methods.

Methods and Regions	Temperature Difference between before and 30 min after Completion of the Transnasal SPGB (°C) (Mean ± SD)		*p*-Value
	Drip-Method Group	Topical-Method Group	
Right forehead area	0.24 ± 0.61	−0.18 ± 0.73	0.001 *
Left forehead area	0.19 ± 0.69	−0.15 ± 0.81	0.015 *
Right maxillary area	0.60 ± 1.21	0.21 ± 1.04	0.046 *
Left maxillary area	0.49 ± 1.08	0.32 ± 1.05	0.353
Right mandibular area	0.43 ± 1.40	0.16 ± 0.83	0.141
Left mandibular area	0.45 ± 1.38	0.16 ± 0.90	0.133
Total	0.40 ± 1.11	0.09 ± 0.91	<0.001 *

* *p*-values less than 0.05 indicate a statistical difference. SPGB, sphenopalatine ganglion block.

**Table 5 jpm-12-00830-t005:** Comparison of VAS scores between drip and topical methods in herpes zoster and migraine.

Disease	Method	VAS before Transnasal SPGB	VAS 30 min after Completion of the Transnasal SPGB	VAS Change	*p*-Value
Herpes zoster	Drip method (55 times)	5.4 ± 1.6	2.9 ± 1.4	−2.6 ± 1.3	<0.001 *
	Topical method (34 times)	5.4 ± 2.0	4.1 ± 1.8	−1.3 ± 1.1	0.008 *
	*p*-value	0.977	0.001 *	<0.001 *	
Migraine	Drip method (4 times)	6.0 ± 1.9	4.1 ± 1.6	−1.9 ± 1.3	<0.001 *
	Topical method (7 times)	6.4 ± 2.0	3.1 ± 1.1	−3.3 ± 1.1	0.004 *
	*p*-value	0.625	0.055	0.014 *	

* *p*-values less than 0.05 indicate a statistical difference. VAS, visual analogue scale; SPGB, sphenopalatine ganglion block.

**Table 6 jpm-12-00830-t006:** Estimation of drip and topical method effect for sympathetic, parasympathetic, sensory nerve, and effective diseases.

Transnasal SPGB	Drip Method	Topical Method
Method	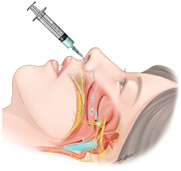	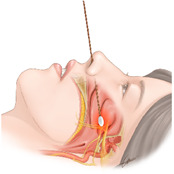
Sympathetic nerve	Block dominantly	Block slightly or exacerbate reactively
Parasympathetic nerve	Block slightly or exacerbate reactively	Block dominantly
Sensory nerve	Block	Block
Dominant effective diseases	Herpes zoster Hyperhidrosis	Migraine Post-dural puncture headache

## Data Availability

Not applicable.

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
