# Peer review of "Comparison of Temperature and Pain Changes between the Drip and Topical Methods of Administering the Transnasal Sphenopalatine Ganglion Block"

_jpm, 2022, doi:10.3390/jpm12050830_

Round 1
Reviewer 1 Report
The aim of the study is interesting and presents itself as a step to better use the SPGB block in different clinical situations. There are some aspects that need to be clarified in order to improve this paper.
1 In the absence of a control group, the data can only be compared between the two methods, and even the efficacy figures are not very generalizable.
2 a systematic collection of adverse events is required
3 How did you calculate the P-value in the row "total" of table 3 and following?
4 data presented in figures 6 - 7 and 8 -9 are already present in tables 3 and 4 respectively; please choose the presentation mode you prefer, tables or figures, not both.
5 The study limits are well reported in lines 288-290. I agree.
6 The conclusion section must be rewritten logically and comprehensively
Author Response
- This research is retrospective study, so we cannot collect control group. Prospective randomized controlled study will be need for control group. And we clean up the ungeneralizable figures.
- There were no systemic adverse events in all patients, Some pain were in topical methods group, and some cough were in drip methods.
- Total means sum of all regions of face. We calculate the mean and SD from sum of all region of face and compared before and after SPGB.
- We choose table 3 and 4 rather than figure 6,7,and 8
- Thank you
- We rewrote conclusion section.
Reviewer 2 Report
Ho Young, and colleagues have submitted a retrospective case-control study describing the different techniques used for the transnasal sphenopalatine ganglion block(SPGB). The authors compared facial temperatures and visual analogue scale (VAS) between the drip method and the topical method of transnasal sphenopalatine ganglion block (SPGB). Previous studies confirmed temperature changes, but no studies have been done to compare changes in temperature between the drip method and topical method. The findings from the current study provide the evidence-based utility of using the drip method for the treatment of facial/head pain associated with herpes zoster infection (which dominantly blocks the sympathetic system) and the topical method for migraine headache and post-dural leak headache (which blocks parasympathetic method). The current study can act as a guide for future studies on this topic.
Great work. I don't have any constructive feedback.
Author Response
Thank you for good comments
Reviewer 3 Report
I find the study very interesting. Here are some comments , which might be helpful to the structure and fluency of this manuscript.
- in the Abstract section: (lines 41 and 42) "before and 30 min.." the world "after" is missing; (line 35 ) specify that VAS score was evaluated only in herpetic neuralgia and migraine (which migraine? with or without aura?).
- in the Methods section: where the inclusion criteria of the selected patients are described, it will be appropriate to make more explicit the diagnoses considered later in the analysis (trigeminal neuralgia, migraine , hyperhidrosis, etc..); in the procedure description (lines 128 -141) there are errors: figure 1A instead of figure 1B;
- in the Results section: "Table 5" is mentioned, but is not attached.
- Within the two treatment groups (drip and topical methods), it would be interesting to compare the temperature and VAS variations, before and after SPGB, in relation to the different diagnoses analyzed. It would be useful to assess the differences , according to the type of pain, in order to better support the discussion, based on the different responses compared to the starting diagnosis.
Author Response
- We wrote missing word “after SPGB”, VAS score was evaluated in herpes zoster, migraine, and atypical facial pain, but was not evaluated in anosmia, tinnitus, and hyperhidrosis. Migraine included both types of migraine with or without.
- We made more explicit the diagnoses. We correct the Figure numbers.
- We attached Table 5.
- Thank you for good comments. We were tried to assess the differences according to the type of pain, but we had insufficient case of various painful or unpainful diseases. We are planning to further research of prospective RCT about SPGB in each disease.
Round 2
Reviewer 3 Report
The manuscript was carefully revised according to previous suggestions.
Tables and figures are clear and understandable.
Author Response
We checked again for english language and styles.